

# Genome-wide characterization and expression analysis of the CONSTANS-like gene family of *Juglans mandshurica* Maxim

Jingwen Wu, Mengmeng Zhang, Yue Gao, Shuhan Li, Ruoxue Jia and Lijie Zhang

Breeding and Cultivation of Liaoning Province, Key Laboratory of Forest Tree Genetics, Shenyang, Liaoning, China
Shenyang Agricultural University, College of Forestry, Shenyang, Liaoning, China

## ABSTRACT

The zinc-finger proteins encoded by the CONSTANS-like (COLs) gene family in *Juglans mandshurica* Maxim. play a significant role in regulating photoperiod-dependent flowering time, as well as in various processes such as growth and development. In this study, 15 members of the CONSTANS-like gene family were identified based on the genomic data of *Juglans mandshurica*. All of these proteins possess an N-terminal zinc-finger B-box domain and a C-terminal CCT domain. Phylogenetic analysis indicates that the *JmCOLs* proteins can be divided into three subgroups, with gene structures and motif compositions varying among these subgroups. Chromosomal analysis reveals that the 15 *JmCOLs* genes are distributed across nine chromosomes. The promoters of genes in this family contain stress-related cis-acting elements, hormone-related response elements, and other elements associated with growth and development. Notably, the most prominent elements are the light-responsive elements, suggesting that genes in this family are predominantly expressed in leaves. The expression patterns of *JmCOLs* genes differ among the members. Specifically, *JmCOL5* and *JmCOL10* are expressed exclusively in flower buds ($p < 0.05$). Throughout the 10 stages of flower bud development, the overall expression level of *JmCOL4a* peaks at approximately 50 to 100 times higher than its lowest point. The expression pattern of *JmCOL5*, which first reaches its maximum during the physiological differentiation stage of protogynous male flower buds before declining, suggests its potential involvement in the development of heteromorphic and dichogamous flowers.

## INTRODUCTION

Flowering serves as a critical indicator of the transition from the vegetative growth phase to the reproductive growth phase, marking a pivotal event in the plant life cycle. The progression of floral development can be systematically categorized into four distinct stages, with the initial stage characterized by the plant's response to both environmental

Corresponding author
Lijie Zhang, Zlj330@syau.edu.cn

and endogenous stimuli (*Song, 2019*). During this stage, various environmental cues, such as light and temperature, along with endogenous signals, including hormones and senescence, can influence and regulate the floral transition. The photoperiodic pathway is one of the primary components controlling flowering and is regulated by light sensors and an internal timer known as the circadian clock. In photoperiodic flowering, CONSTANS (CO) plays a crucial role as a key member of the photoperiodic pathway, serving as the central hub that integrates all upstream signals and activates the expression of the flower-forming gene FT (flowering locus T) in the leaf (*Imaizumi et al., 2005*). Then, FT proteins can move from the leaf to the apical meristem through the vascular bundle, triggering the program that transforms it into a reproductive meristem. Ultimately, these changes induce flower production (*Romero & Valverde, 2009*). The central gene of the photoperiodic pathway is CONSTANS (CO), which serves as the integrator of day length signals and the output of the circadian clock (*Lv et al., 2021*). CONSTANS-like proteins are a class of zinc finger transcription factors that are conserved across plants. The N-terminus of these proteins contains one or more zinc finger B-box domains, characterized by the sequence of two cysteines (C-X2-C) followed by a stretch of 16 variable amino acids (X) and concluding with another two cysteines (C-X2-C). These domains are crucial for protein-protein interactions. The C-terminus features a 43-amino-acid domain known as the CCT (CONSTANS, CONSTANS-like, TOC1) structural domain, which plays a significant role in nuclear localization signaling and mediates interactions with DNA and various proteins (*Xing et al., 2022*).

Research has demonstrated that the regulation of COL genes during flowering is conserved among different plant species; however, the transmission pathways may vary among different members. Some long-day plants must accurately detect changes in day length to flower at the right time, ensuring reproduction during the appropriate season. In contrast, short-day plants possess a distinct photoperiod response mechanism. This evolutionary divergence, driven by ecological adaptability, results in variations among COL family members across different species, allowing them to fulfill their specific physiological requirements. The total number of COL family members is unevenly distributed among various species. For instance, in long-day plants, there are 16 members in *rice*, nine in *Hordeum vulgare* (*Griffiths et al., 2003a*), 10 in *Beta vulgaris* (*Dally et al., 2018*), 26 in *Glycine max* (*Wu et al., 2014*), 20 in Raphanus sativus (*Liu et al., 2020*). In day-neutral plants, there are also differences in the number of gene members: 25 in *Brassica campestris* (*Song et al., 2015*), 11 in *Chrysanthemum lavandulifolium* (*Fu, Yang & Dai, 2015*), 25 in banana (*Musa acuminata*) (*Chen et al., 2012*), and 12 in grape (*Vitis vinifera*) (*Hu et al., 2018*). However, in the short-day plant *Gossypium hirsutum*, the number of COL family members is as high as 42 (*Han et al., 2022*).

In the model plant *Arabidopsis thaliana*, 17 CONSTANS-like (COL) genes have been identified. The COL gene family plays crucial roles in various developmental processes, including flowering time, branching, shade avoidance responses, photomorphogenesis, and embryonic axis elongation. These COL family members have been classified into three main groups: the first group contains two B-box structural domains, a CCT structural domain, and a VP motif (the valine-proline motif responsible for interaction with COP1).

Class II includes one B-box and one CCT structural domain, while Class III consists of one B-box, a divergent zinc finger structural domain, and a CCT motif (*Yang et al., 2024*).

The regulation of COL genes during flowering is conserved across various plant species; however, the regulatory mechanisms may differ among the different gene members. For instance, in *A. thaliana*, *AtCO* and *AtCOL5* promote flowering under long-day (LD) and short-day (SD) conditions, respectively. In contrast, *AtCOL3* and *AtCOL4* act as flowering repressors in both LD and SD conditions, while *AtCOL8* and *AtCOL9* delay flowering exclusively under LD conditions. Under LD conditions, *AtCOL3* (BBX4) and *AtBBX32* interact at the N-terminus to target FT, thereby regulating flowering (*Steinbach, 2019*). In rice, OsCO3 serves as a specific flowering repressor under SD conditions, while *OsCOL4*, *OsCOL9*, *OsCOL10*, *OsCOL13*, *OsCOL15*, and *OsCOL16* negatively regulate flowering in both LD and SD conditions (*Kim et al., 2008*). In barley, *HvCO1* induced flowering under LD and SD conditions, however *HvCO9* inhibited flowering under SD conditions. In addition, the involvement of *CO/COL* genes in the regulation of flowering has been found in potato (*Solanum tuberosum*), sorghum (*Sorghum bicolor*), sugar beet (*Beta vulgaris*) and bamboo (*Phyllostachys heterocycla*) (*Campoli et al., 2012*).

In addition to regulating flowering, COL genes play significant roles in various aspects of plant growth and development. *AtCOL1* and *AtCOL2* in *A. thaliana*, *PnCOL1* in *Ipomoea nil* (L.). *GmCOL10* in *Glycine max* have been identified as key regulators of the circadian clock (*Wu et al., 2014*). Furthermore, *AtCOL3* in *A. thaliana* has been shown to promote lateral root development and aboveground branching, while *AtCOL4* has been found to enhance tolerance to abiotic stresses (*Steinbach, 2019*). The *Solanum tuberosum StCO* gene has been demonstrated to play a role in photoperiodic tuber formation (*González-Schain et al., 2012*). In *M. acuminata*, *MaCOL1* may be involved in stress response and fruit ripening (*Chen et al., 2012*). In *Chlamydomonas reinhardtii*, CO genes are involved in regulating lipid synthesis (*Deng et al., 2015*). In *V. vinifera* buds, *VvCO* and *VvCOL1* induce flowering and dormancy, and CO homologous genes are expressed in grapevine tendrils, suggesting a possible association with tendril development (*Almada et al., 2009*).

*Juglans mandshurica* Maxim is a deciduous tree belonging to the genus *Julans* of the *Juglandaceae* family. It is a precious timber tree species, a woody grain and oil tree species, and an economic forest tree species for fruit and timber. It combines economic value, nutritional value and medicinal value. Mainly distributed in Japan, North Korea, the Russian Far East and northeastern China, with broad prospects for development and utilization (*Wang et al., 2019*). Under natural conditions, flower of *J. mandshurica* is monoecious and exhibits a high degree of morphodifferentiation in color, size, and morphology. Male flowers are in catkins, while four to ten female flowers form a spike (Fig. 1). Flowering in *J. mandshurica* can in general be grouped into two clear mating types, protogynous and protandrous, and these are randomly distributed in specific populations. However, the development of female and male flowers is not synchronous in *J. mandshurica*, with different flowering periods in the same plant, and the maturation of male and female flowers is sequential and very stable among different plants, which is a reproductive characteristic that improves the heterosis rate and the quantity and quality of fruits of *J. mandshurica* by natural pollination. At present, the molecular mechanism

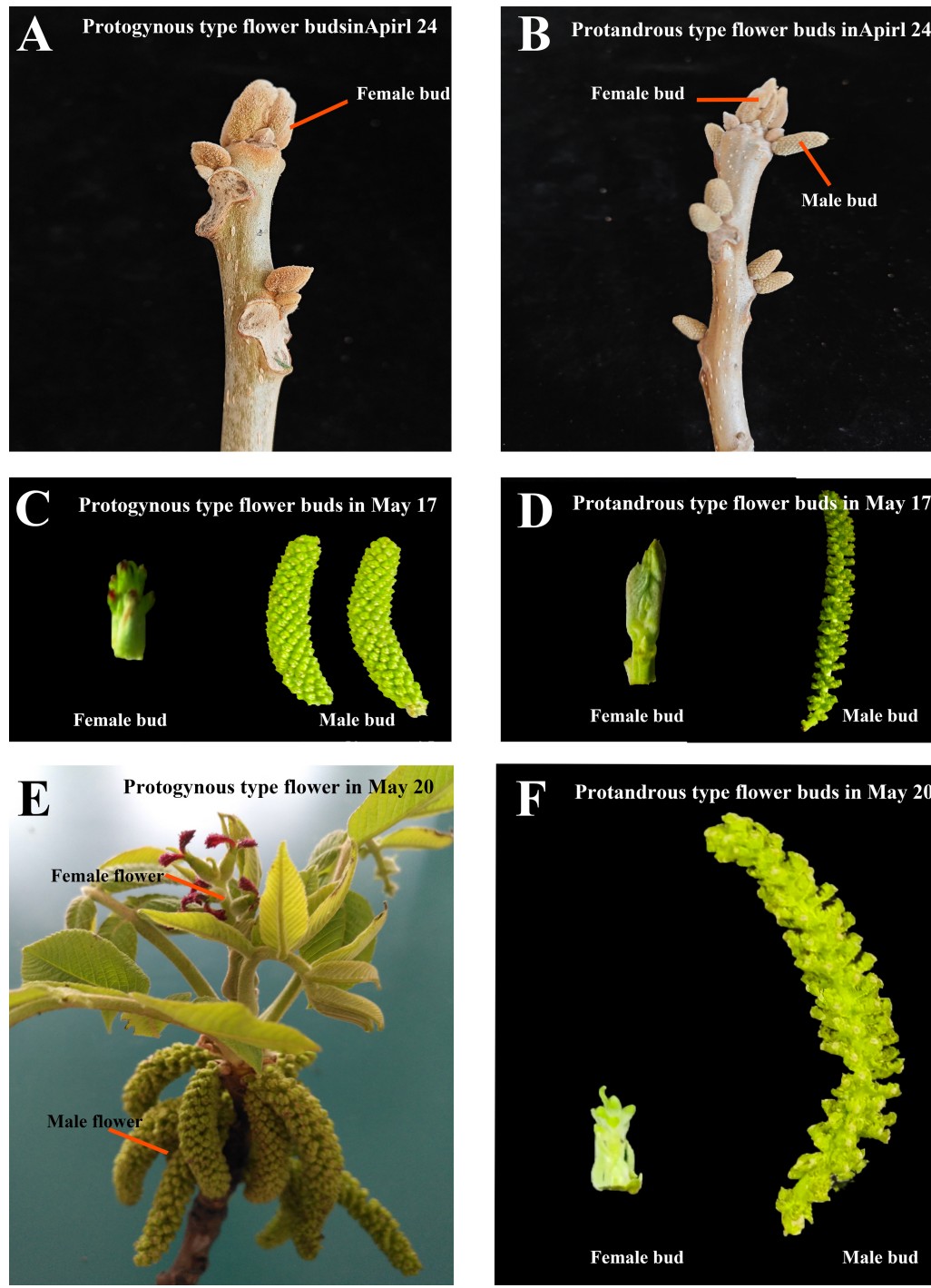

**Figure 1    Protogynous and protandrous type flower buds in different periods.** (A–B) The protogynous and protandrous flower buds at the physiological differentiation stage on April 24. (C–D) The protogynous and protandrous flower buds at the morphological differentiation stage on May 17. (E–F) The protogynous mature flower and protandrous flower buds on May 20.

of bud development and flower formation in *J. mandshurica* has not been fully analyzed (*Zhang et al., 2021*).

Due to the special flower bud development pattern of *J. mandshurica*, the exploration of its molecular mechanism of flower development can provide a theoretical basis for its reproductive growth rules. This paper aims to further explore the potential role of the CONSTANS (CO) gene family in the mechanism of flower bud sex differentiation of *J. mandshurica* in the pathway of photoperiod regulating flower development through bioinformatics analysis, phylogenetic analysis, and expression analysis in different tissues and different types of flower buds of the CONSTANS-like family.

## MATERIALS & METHODS

### Identification of *JmCOL* genes in *Juglans mandshurica*

In the phylogenetic tree, the protein sequences of the CONSTANS (CO) gene family in *A. thaliana* were obtained through TAIR (https://www.arabidopsis.org/) (*Yang et al., 2024*). *A. thaliana* CO protein sequences were used to search the *J. mandshurica* genome database using the blastx program in BLAST (blast-2.6.0+) with $E$-value $< 10^{-10}$. The HMM (Hidden Markov Model) of COLs was constructed by adopting HMMER 3.3.2 (http://www.hmmer.org/download.html). The results of BLAST and HMMER were merged, and the incomplete or redundant sequences were eliminated. A total of 15 *JmCOL* genes with both B-BOX and CCT conserved domains of *J. mandshurica* were identified. In addition, the information and various physicochemical parameters of *JmCOL* genes were analyzed using ProtParam (https://web.expasy.org/protparam/) (*Yang et al., 2024*).

### Chromosomal localization of *JmCOLs* genes, analysis of gene structure and conserved domains

*J. mandshurica* exons and introns of *JmCOL* genes were displayed using GSDS2.0 (http://gsds.gao-lab.org/). Conserved motifs in *JmCOL* were analyzed using MEME software (https://meme-suite.org/meme/tools/tomtom) with default parameters, as previously described (*Yang et al., 2024*). The JmCOL gene was localized on chromosomes using TBtools-II (Toolbox for Biologists) version 2.136. The conserved domains identified through MEME analysis were also marked using TBtools-II (Toolbox for Biologists) version 2.136.

### *JmCOLs* phylogenetic analysis

Phylogenetic analyses were generated with MEGA7.0.26 software. Sequence alignment of proteins from two species (*J. mandshurica*, *A. thaliana*) was performed with MUSCLE, and a phylogenetic tree was constructed with the Neighbor-Joining (NJ) tree, Bootstrap replications = 5 00, Method is p-distance model, Site Coverage Cutoff 50%. The phylogenetic tree constructed by MEGA 7.0.26 was uploaded to iTOL (https://itol.embl.de/) for further editing.

## Collinearity analysis

Genome data of *J. mandshurica* and *A. thaliana* were used to analyze their collinearity relationships. TBtools-II v2.136 software was used to calculate and draw the whole gene collinearity analysis of species.

## Cis-acting elements analysis

The 2 kb DNA sequence upstream from the ATG of JmCOL genes was extracted from the *J. mandshurica* genome database (https://cmb.bnu.edu.cn/juglans/) as the promoter region. Cis-elements were analyzed in the PlantCARE database (http://bioinformatics.psb.ugent.be/webtools/plantcare/html/). Analysis results were visualized with TBtools-II v2.136 software.

## Real-time quantitative PCR (RT-qPCR) analysis

Total RNA was extracted from various organs and tissues of the *J. mandshurica* using a plant RNA extraction mini kit. Its integrity and concentration were then assessed through agarose gel electrophoresis and measured with a Thermo NanoDrop 2000 instrument; cDNA was synthesized using HyperScriptTM III RT SuperMix for qPCR with gDNA Remover to synthesize the first strand of cDNA. Quantitative primers were designed using PrimerPremier6 software, and primer specificity was verified using Primer-BLAST (http://www.ncbi.nlm.nih.gov/tools/primer-blast/) from NCBI, with 18S as the internal reference gene. The instrument used for Real-time fluorescence quantitative PCR is from Analytik Jena, and its name and model are 3107B - 0545 and qTOWER3/G respectively. The reaction system for quantitative PCR has a volume of 20 µL. In this system, the quantitative enzyme used is Taq Pro Universal SYBR qPCR Master Mix. The total amount of cDNA in the reaction volume is 0.1–1 µM. The forward and reverse quantitative primers each have a volume of 0.4 µL (10 µM). Finally, ddH$_2$O is added to make up the volume to 20 µL. The instruments and the reaction procedures used are detailed in Table S2.

## Plant materials

On April 2, 2022, in the natural mother forest of Liao Ning province experimental forestry field, Liaoning, China (124.9°E, 42.13°N), healthy protogynous and protandrous type of *J. mandshurica* were chosen. At this time, *J. mandshurica* buds are at the stage of physiological differentiation, that is, the period when flower buds are just produced as shown in Figs. 1A–1B. Four bud types were collected with three replicate samples for each group of bud types and numbered one by one. The samples in group T1 were protogynous female buds named CC (A1, A2, A3), the samples in group T2 were protogynous male buds named CX (A4, A5, A6), the samples in group T3 were protandrous female buds named XC (B1, B2, B3), and the samples in group T4 were protandrous male buds named XX (B4, B5, B6). The total number of samples was 12. Flower buds were rapidly frozen in liquid nitrogen and stored at −80 °C. A total of 12 RNA seq libraries (A1-6, B1-6) were constructed. The collected samples were sent to Paisano Biotechnology Ltd (Shanghai, China) for high-throughput sequencing of the cDNA libraries. The project number is YFnj20201682. The sequencing data were filtered using Cut adapt to remove any junctions at the 3′ end and any reads with an average mass fraction lower than Q20, in order to obtain clean reads for subsequent analysis and identification.

Starting from April 2, 2024, samples were collected from ten different developmental periods: April 2 (4.2), April 10 (4.10), April 14 (4.14), April 24 (4.24), April 30 (4.30), May 4 (5.4), May 13 (5.13), May 17 (5.17), May 20 (5.20), and May 22 (5.22). The period from April 2 to April 30 was characterized by physiological differentiation, while the latter half of this period was marked by morphological differentiation. Protogynous and protandrous type flower buds from the natural seed forest of *J. mandshurica* at the experimental forestry farm in Liaoning Province. Four different types of flower buds were collected (Figs. 1A–1F) with each sample containing three independent biological replicates. The number of samples per period was 12 (each of the four types of buds contained three biological replicates). The total number of all samples was 120, and these flower bud samples were quickly frozen in liquid nitrogen after collection and stored at −80 °C refrigerator.

On June 2, 2024, at the experimental base of Shenyang Agricultural University, an appropriate quantity of samples was carefully collected from the leaves, flower buds, stems, and fruits of *J. mandshurica*. Each sample type underwent three biological replicates, summing up to a total of 12 samples. Post-collection, these samples were promptly wrapped in aluminum foil and plunged into liquid nitrogen for rapid freezing. Subsequently, they were transferred and stored in an ultra-low-temperature refrigerator set at −80 °C for future experimental requirements.

## Data and statistical analysis

The *J. mandshurica* genome database (Genome Assembly V1.3) was obtained from the website (https://cmb.bnu.edu.cn/juglans/) (*Mortazavi et al., 2008*). To facilitate the comparability of gene expression levels across genes and samples, the expression amounts were normalized using FPKM (Fragments Per Kilobase of transcript per Million map pe dreads) (*Trapnell et al., 2010*). FPKM is calculated as the number of fragments per kilobase of gene length per million mapped fragments, based on the alignment results. The heatmap was plotted by https://www.bioinformatics.com.cn (last accessed on 10 Oct 2024), an online platform for data analysis and visualization (*Tang et al., 2023*).

All experimental data included at least three biological replicates. The tissue-specific expression maps of *JmCOLs* gene were performed using Excel software (Microsoft Office, 2021). Bar charts were determined by one-way ANOVA (* stands for $p < 0.05, 0.01$). The relative expression of nine *JmCOL* genes was calculated by $2^{-\Delta\Delta CT}$ method (*Livak & Schmittgen, 2001*), and the results were subjected to one-way ANOVA (* stands for $p < 0.05, 0.01$) using SPSS 26. *Post hoc* multiple tests were performed on the data results, assuming that the equal variance was Duncan H. The tissue-specific expression maps of *JmCOLs* gene were performed using Microsoft Excel (Microsoft, Inc., Redmond, WA, USA).

**Table 1** *JmCOL* gene family members and physicochemical properties of their encoded proteins in *Juglans mandshurica*.

| Gene name | Locus ID | CDS length/bp | Amino acid length/aa | Molecular weight (kDa) | Isoelectronic point | Instability index | Subcellular localization |
|---|---|---|---|---|---|---|---|
| *JmCOL2* | JMA012135 | 1,119 | 372 | 41.22 | 5.30 | 44.47 | Golgi body |
| *JmCOL4a* | JMA031564 | 1,026 | 341 | 37.57 | 6.59 | 51.70 | Golgi apparatus, Nucleus |
| *JmCOL4b* | JMA004774 | 1,038 | 345 | 37.94 | 6.16 | 47.32 | Golgi apparatus, Nucleus |
| *JmCOL5* | JMA008296 | 1,092 | 363 | 39.63 | 5.83 | 46.47 | Golgi apparatus, Nucleus |
| *JmCOL6* | JMA020086 | 1,398 | 465 | 52.51 | 5.22 | 37.42 | Golgi apparatus, Nucleus |
| *JmCOL9a* | JMA004947 | 1,251 | 416 | 45.03 | 5.18 | 37.39 | Golgi apparatus |
| *JmCOL9b* | JMA029849 | 1,551 | 516 | 57.69 | 5.61 | 42.10 | Golgi apparatus, Nucleus |
| *JmCOL10* | JMA026904 | 1,557 | 518 | 57.54 | 5.70 | 43.51 | Golgi apparatus, Nucleus |
| *JmCOL11* | JMA000719 | 1,263 | 420 | 45.71 | 5.24 | 51.58 | Nucleus |
| *JmCOL12* | JMA001149 | 1,287 | 428 | 47.18 | 5.26 | 37.35 | Golgi apparatus |
| *JmCOL13* | JMA026099 | 1,095 | 364 | 40.85 | 5.44 | 42.61 | Golgi apparatus |
| *JmCOL14* | JMA017943 | 1,428 | 475 | 51.81 | 5.77 | 41.39 | Golgi apparatus |
| *JmCOL15* | JMA022230 | 1,440 | 479 | 52.47 | 6.68 | 38.40 | Golgi apparatus |
| *JmCOL16a* | JMA033005 | 1,215 | 404 | 45.87 | 5.59 | 44.30 | Golgi apparatus |
| *JmCOL16b* | JMA019919 | 1,332 | 443 | 48.82 | 5.79 | 37.00 | Golgi apparatus |

## RESULTS

### Identification and analysis of the *JmCOL*s gene family in *Juglans mandshurica*

The structural domain integrity of the candidate members was confirmed using NCBI Batch CD-search and the MEME website (https://meme-suite.org/meme/tools/tomtom). Ultimately, the protein sequences encoded by 15 *JmCOL* genes were obtained, and each gene was named according to its level of homology to A. thaliana *CO* and *COLs* (Table 1). It was observed that the full length of the CDS sequences of the *JmCOLs* genes ranged from 1,026 bp to 1,557 bp, encoded 341–518 amino acids, and had molecular weights of 37.57–52.5 kDa. Additionally, the theoretical isoelectric point ranged from 5.18 to 6.68, indicating that all of the *JmCOL* proteins were acidic. Subcellular localization prediction revealed that the majority of *JmCOL* proteins are localized to the nucleus, while the remaining members are localized to the Golgi apparatus.

The chromosomal localization of *JmCOL* genes was conducted, revealing that 15 of these genes are distributed across nine chromosomes of the *J. mandshurica* genome. This distribution is illustrated in Fig. 2. For example, *JmCOL 2*, *JmCOL12*, and *JmCOL9a* were located on chromosome 3; *JmCOL 13*, *JmCOL5*, and *JmCOL4a* were located on chromosome 7; and *JmCOL11*, *JmCOL4b*, *JmCOL16b*, *JmCOL6*, and *JmCOL16a* were located on chromosomes 4, 8, 9, 16, and 15, respectively. Chromosome 5 and chromosome 6 each contain two genes.

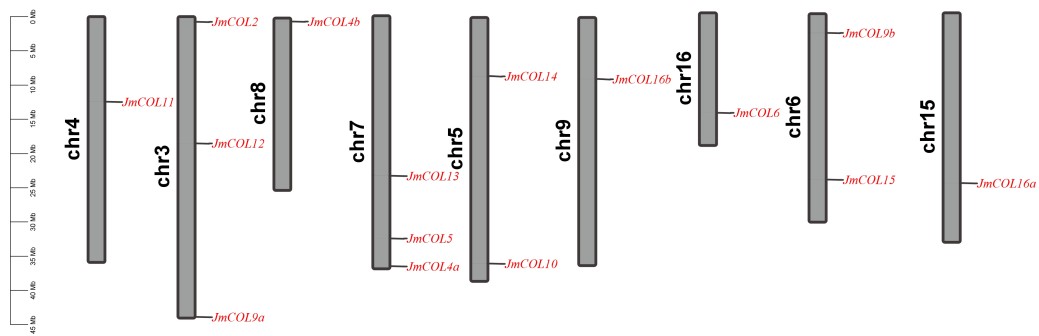

**Figure 2**  Chromosomal localization of *JmCOL* genes in *Juglans mandshurica*.

## Phylogenetic analysis of the *JmCOL*s gene family in *Juglans mandshurica*

Phylogenetic and structural diversity play a key role in the evolutionary relationships of gene families. COL protein sequences were downloaded from the *A. thaliana* database to construct a phylogenetic tree of COL proteins in nutmeg and *A. thaliana* (Fig. 3). The *CO* family can be divided into three main taxa in *A. thaliana*: Taxon 1 includes *CO* and *COL1~COL5* with two B-boxes and six conserved amino acids known as T motifs at their carboxyl termini; taxon 2 includes *COL6~COL8* and *COL16* that have one B-box; and Taxon 3 includes *COL9~COL15*, with one B-box and a second divergent zinc finger structural domain (*Hassidim et al., 2009*). Based on the evolutionary tree, *JmCOL* proteins can be classified into three groups (Group I, Group II and Group III), where Group III contains the most *JmCOL* family members with eight, followed by Group I with four and Group II with the least number of three members.

## Conserved motif and gene structure analysis of the *JmCOL*s protein family in *Juglans mandshurica*

To further investigate the structural diversity of *JmCOL* genes, we compared the number and composition of exons and introns of *JmCOLs* genes, as well as the corresponding evolutionary relationships (Fig. 4A), and the results were similar to those in Fig. 3. Gene structure analysis showed that *JmCOL* proteins of the same subgroup usually shared similar exons and introns. However, the number of exons in a subgroup also varies.

Conserved motif prediction of *JmCOL* proteins using the MEME online website yielded 10 conserved motifs (Fig. 4B), and the motif types, numbers, and positions of the same branch genes were more consistent. The conserved motifs of COL proteins were further analyzed using the MEME software, and a total of 10 conserved motifs were identified and designated as motifs 1–10 (Fig. 4C). All genes contained motif 1 (CCT structural domain) and motif 2 (B-box structural domain). Furthermore, similar to evolutionary trees and gene structures, proteins with the same conserved structural domains tend to belong to the same subgroup.

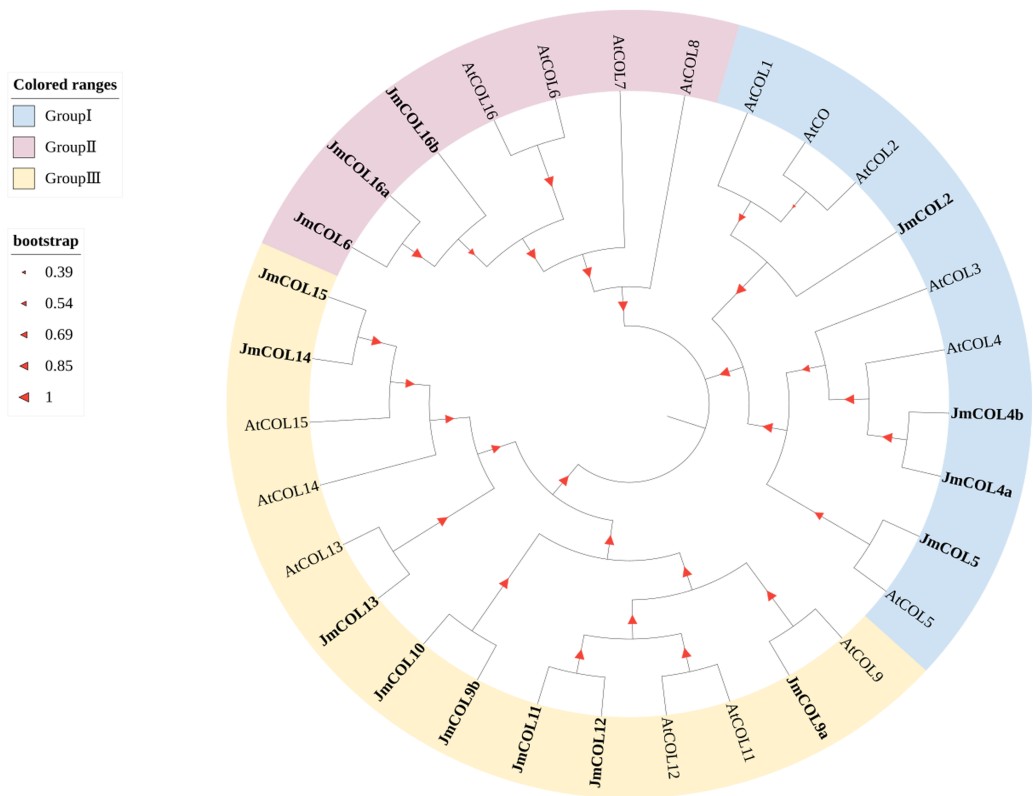

**Figure 3** Phylogenetic analysis of COL proteins sequences in *Juglans mandshurica* (15 *Jm COLs*), *Arabidopsis* (16 *AtCOLs*). The *JmCOLs* gene were labeled in bold, and the different groups were overall labeled in light red, blue and yellow, respectively. The size of the red triangle positively represents the homology level.

### *JmCOL*s promoters involved a large number of light responsiveness elements

In order to analyze the functional differences among the various members of the *J. mandshurica JmCOL* gene family, the promoters of the family genes were analyzed for cis-acting elements (Table 2). The results indicated that the walnut *JmCOLs* family genes possess eight light-response-related cis-elements, seven hormone-related cis-elements, five stress-related cis-elements, and cis-elements responsive to other growth and developmental processes. Among the 11 *JmCOLs* genes that contained BOX-4, a cis-acting element involved in light response, eight genes also contained ARE, a cis-acting regulator necessary for anaerobic induction. Additionally, eight genes contained CGTCA-motif, a cis-acting regulator involved in the response to MeJA. These findings suggest that the *JmCOLs* may be responsive to light, anaerobic induction, and MeJA. Furthermore, the prevalence of light-responsive related cis-elements indicates that the *JmCOL* family contributes significantly to light response and, to a lesser extent, potentially to hormone-related physiological processes.

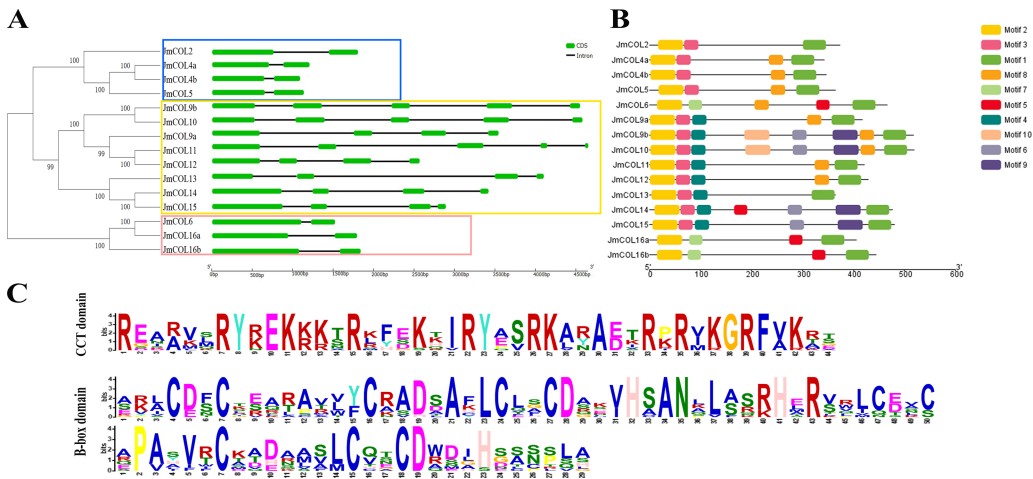

**Figure 4** **Analysis of *JmCOL* protein evolution, conserved motif (A), conserved domain (B) and conserved domain sequence (C).** The different three gene groups were marked by three different colored rectangular box.

## Comparative genome collinearity analyses of COL genes between *Juglans mandshurica* and *Arabidopsis thaliana*

The distribution of the 15 *JmCOL* genes on the *J. mandshurica* chromosomes was analyzed based on the genomic information available for this species. As illustrated in Fig. 5A, gene duplication events were identified for six of these gene pairs, all of which were segmental duplications. To investigate the evolutionary relationship of *COL* genes, a covariance analysis was conducted on both *J. mandshurica* and *A. thaliana*. The results demonstrated that a total of 18 pairs of genes exhibited homology between the *J. mandshurica* and *A. thaliana* (Fig. 5B).

## *JmCOL*s showed a tissue-specific expression in different tissues

In order to explore the expression pattern of *JmCOL*s genes in different tissues of *J. mandshurica*, we first studied the expression patterns of *JmCOL* family members in different tissues such as stems, fruits, leaves and flower buds by qRT-PCR. The effective experimental results of nine genes were obtained (Fig. 6), and these genes showed several different expression patterns. *JmCOL 2*, *JmCOL4a*, *JmCOL 6* and *JmCOL13* were mainly expressed in leaves. The transcription levels of *JmCOL2* and *JmCOL6* in leaves were at least two times that of other tissues. *JmCOL16 a* was mainly expressed in leaves, and the expression level was 14–23 times that of other tissues. *JmCOL5* and *JmCOL10* were abundantly expressed in fruits and flower buds, while *JmCOL6*, *JmCOL13*, *JmCOL14* and *JmCOL16 a* were lowly expressed in flower buds. *JmCOL4a* was highly expressed in flower buds, fruits and leaves. Among the expression of *JmCOL9a* in different tissues, the highest expression level in stems was three times that in leaves, and this expression pattern was quite different from other members. These results indicate that the *JmCOL*s family may play an important role in all aspects of the growth and development of *J. mandshurica*, and several members with similar expression patterns may have similar functions.

**Table 2  Prediction of cis-acting elements on the promoters (2,000 bp upstream of the gene) of *JmCOLs*.**

| Cis-element | Gene number | Sequence | Function | Cis-element type |
|---|---|---|---|---|
| Box 4 | 11 | ATTAAT | part of a conserved DNA module involved in light responsiveness | |
| TCT-motif | 8 | TCTTAC | part of a light responsive element | |
| G-Box | 5 | CACGTG | cis-acting regulatory element involved in light responsiveness | |
| GT1-motif | 3 | GGTTAA | light responsive element | |
| AE-box | 3 | AGAAACAA | part of a module for light response | Light responsive |
| G-box | 2 | CACGAC | is-acting regulatory element involved in light responsiveness | |
| GATA-motif | 1 | GATAGGA | part of a light responsive element | |
| GA-motif | 1 | ATAGATAA | part of a light responsive element | |
| LTR | 2 | CCGAAA | cis-acting element involved in low-temperature responsiveness | |
| CCAAT-box | 3 | CAACGG | MYBHv1 binding site | |
| MBS | 2 | CAACTG | MYB binding site involved in drought-inducibility | Abiotic and biotic stresses |
| ARE | 8 | AAACCA | MYB binding site involved in drought-inducibility | |
| HD-Zip 1 | 1 | CAAT(A/T)ATTG | element involved in differentiation of the palisade mesophyll cells | |
| TATA-box | 14 | TATA | core promoter element around -30 of transcription start | |
| A-box | 1 | CCGTCC | cis-acting regulatory element | Other growth and development related elements |
| CAAT-box | 14 | CAAT | MYBHv1 binding site | |
| GARE-motif | 1 | TCTGTTG | gibberellin-responsive element | |
| TGACG-motif | 2 | TGACG | cis-acting regulatory element involved in the MeJA-responsiveness | |
| TGA-element | 2 | AACGAC | auxin-responsive element | |
| TATC-box | 2 | TATCCCA | cis-acting element involved in gibberellin-responsiveness | |
| CGTCA-motif | 8 | CGTCA | cis-acting regulatory element involved in the MeJA-responsiveness | Phytohormone responsive |
| P-box | 3 | CCTTTTG | gibberellin-responsive element | |
| AuxRR-core | 1 | GGTCCAT | cis-acting regulatory element involved in auxin responsiveness | |

## Expression profiling of *JmCOL*s from transcriptome data

In the 12 physiological differentiation stage flower bud samples of *J. mandshurica* transcriptome, the gene expression coverage of each sample was shown in the form of density map. The FPKM density distribution map (Fig. S1) provides an overview of all gene expression patterns in the sample. In general, moderately expressed genes account for the majority, and a small number of genes are expressed at low and high levels. The FPKM value is proportional to the amount of gene expression. It can be seen from the graph that most of the 12 groups of samples overlap with each other. As shown in the attached figure, most of the intervals of the 12 groups of samples overlap with each other, indicating that the expression levels of the genes before and after treatment are similar, and only a very

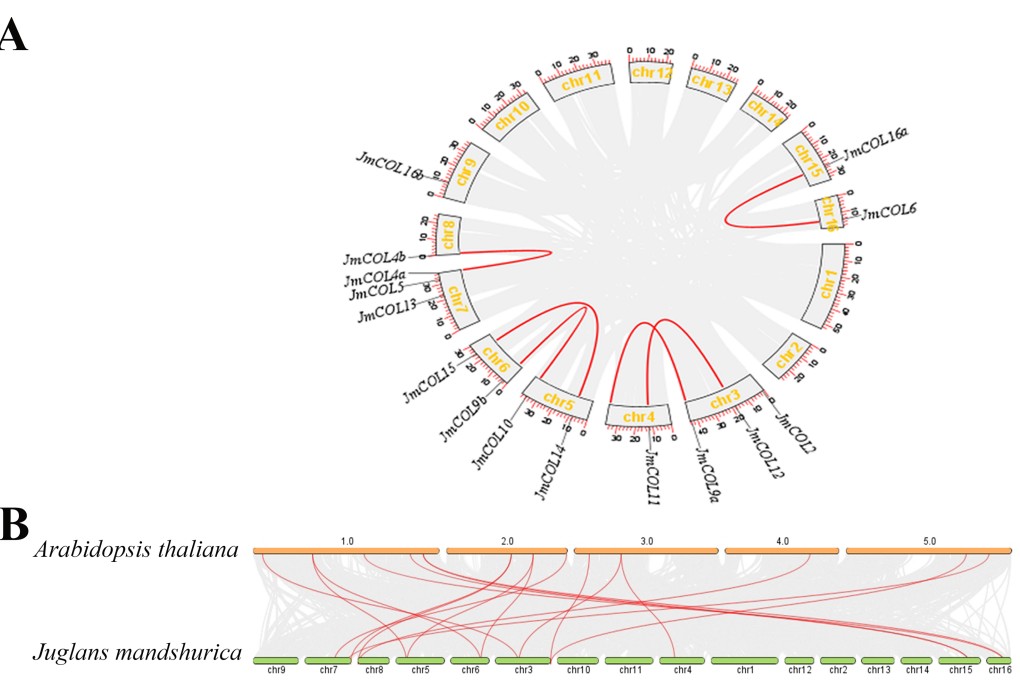

**Figure 5** **Analysis of the replication pattern of the *COLs* gene in *Juglans mandshurica*.** (A) Collinearity analyses of *JmCOL* genes. Red lines represent the syntenic relationships of COLs and gray lines represent the syntenic relationships in the genome of *J. mandshurica*. (B) Comparative genome collinearity analyses of COL genes between *A. thaliana* and *J. mandshurica*. Red lines represent the syntenic relationships of COLs and gray lines represent the syntenic relationships between genome wide of *A. thaliana* and *J. mandshurica*.

small part is different. The non-overlapping part may be the change of gene expression level caused by heterologous mechanism. In a heat map of FPKM values plotted for the *JmCOLs* genes, it can be seen that most members of the *JmCOLs* gene family are expressed in both male and female buds at the time of physiological differentiation, but the high expression of *JmCOL5*, *JmCOL4a*, *JmCOL9b*, and *JmCOL11* in the type A samples implies that the female-preferred buds have a higher specificity of expression.

In the heat map of FPKM (Fig. 7A) values plotted for *JmCOLs* genes, it was found that most of the members of the *JmCOLs* gene family were expressed in both male and female flower buds during the physiological differentiation period, but the specific high expression of *JmCOL5*, *JmCOL4a*, *JmCOL* 9b, and *JmCOL* 11 in the type A samples revealed their possible involvement in protogynous bud formation and development.

At the physiological differentiation stage, correlation analysis based on the FPKM values of the transcriptome data was carried out with SPSS 26 and heat mapping was performed with TBtools. From the correlation analysis heatmap (Figs. 7B–7D), there were a large number of significant positive correlations in the expression of the *JmCOLs* gene family, (Fig. 7B). Notably there were also significant negative correlations between *JmCOL4b*/*JmCOL9b* in protandrous type of flower buds (Fig. 7C) and *JmCOL9b/15* in protogynous type flower buds (Fig. 7B), and between the two types of flower buds,

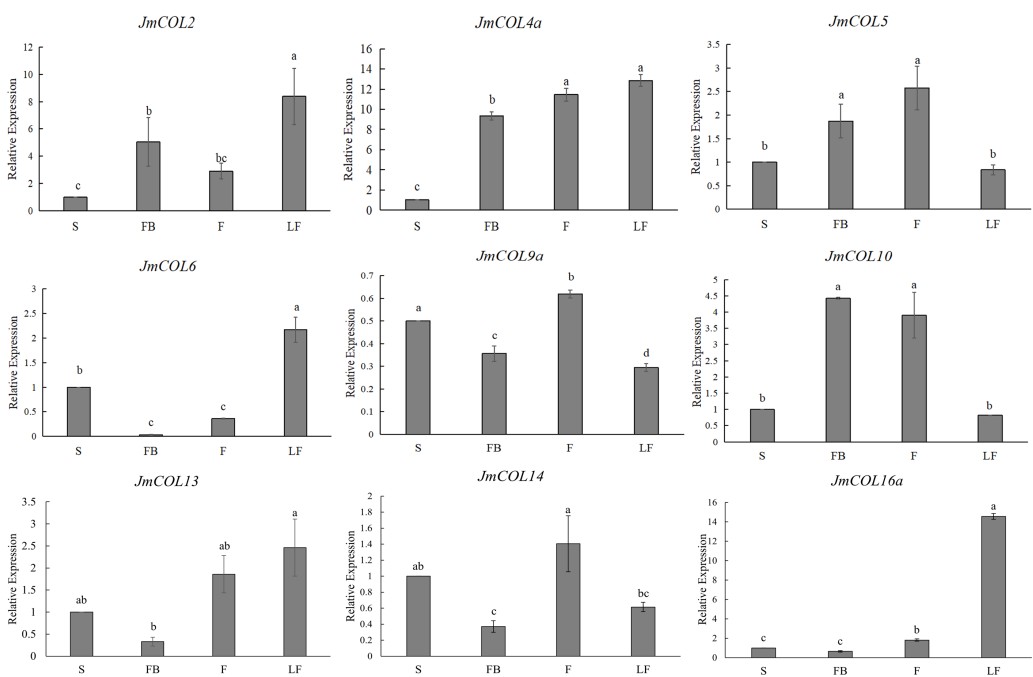

**Figure 6** **Expression profiles of nine members of the *JmCOLs* family in different tissues of *Juglans mandshurica*.** The tissue development stages used for analysis were: S (stem), FB (flower bud), F (fruit), LF (leaf). Values represent the mean ± standard deviation of three biological replicates. Different lowercase letters indicate significant differences between treatments ($p < 0.05$), the same below.

*JmCOL5/16a*, and *JmCOL9b/14/15* in both flower buds (Fig. 7D). The specificity and correlations in the expression of *JmCOL*s reveal the potential involvement of members of this family in the process of bud sex differentiation during the early stages of bud differentiation.

## Different types of flower buds and developmental periods have significant effects on the expression of *JmCOL4a*/5

According to the results of the heatmap, *JmCOL4a*/5 embodied specific expression in protogynous flower buds during the early stages of bud differentiation, and there were some positive and negative correlations with other genes. In order to see whether the expression of these two genes throughout bud differentiation is significantly differentiated among different types of buds, and thus to further validate the potential participation of the *JmCOL* gene family in the bud differentiation of huckleberry, we selected bud samples from ten developmental periods: April 2 (4.2), April 10 (4.10), April 14 (4.14), April 24 (4.24), April 30 (4.30), May 4 (5.4), May 13 (5.13), May 17 (5.17), May 20 (5.20), and May 22 (5.22) and performed qRT-Probe assays for the *JmCOL4a*/5 levels of *JmCOL4a*/5 were assayed by qRT-PCR. Statistics were plotted in Excel by one-way and multifactorial ANOVA in SPSS26 for s four types of flower buds (CC, CX, XC, XX) at different time periods.

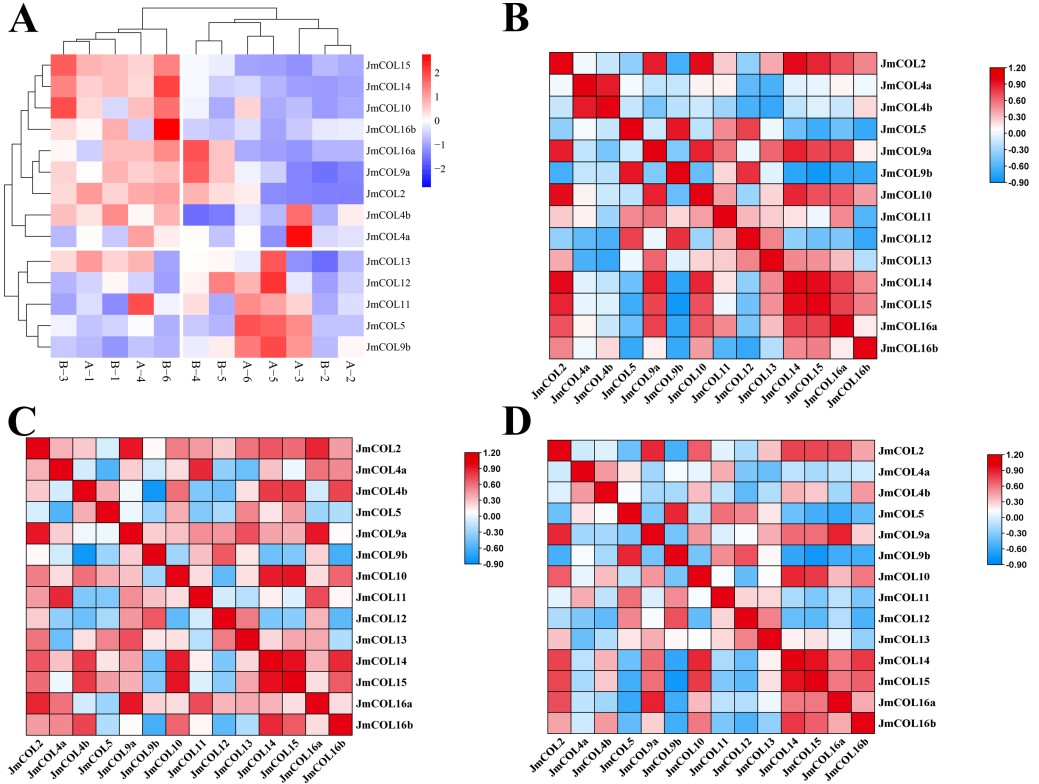

**Figure 7** **The expression and correlations of 15 *JmCOLs* in flower buds of *Juglans mandshurica*.** (A) Expression patterns of *JmCOLs* in flower buds in physiological differentiation period. From color blue to red, the expression level increased from low to high. The flower bud type of A1-3 was protogynous female buds named CC, the flower bud type of A4-6 was protogynous male buds named CX, the flower bud type of B1-3 was protandrous female buds named XC, and the flower bud type of B4-6 was protandrous male buds named XX. (B). Analyses of correlation between transcript levels of JmCOLs in protogynous flower buds. (C) Analyses of correlation between transcript levels of JmCOLs in protandrous flower buds. (D) Analyses of correlation between transcript levels of JmCOLs in protogynous and protandrous flower buds.

Based on the results of statistical analysis, it was known (Table S5) that the expression of *JmCOL4a*/5 was significantly affected by the type of flower bud and the period of flower bud development ($p < 0.05$). *JmCOL4a* was lowly expressed during the physiological differentiation phase stage (April 2 to April 30), whereas it showed a trend higher than the physiological differentiation phase during the morphological differentiation phase (May 4-May 22) (Fig. 8). In terms of expression at different periods (Table S5), the expression of *JmCOL4a* in all four types of flower buds reached its highest value on May 20, with the expression being approximately 50–100-fold higher than that on April 14, the period with the lowest expression, respectively. From the viewpoint of flower bud types, *JmCOL4a* expression was overall higher in male than in female flower buds (Fig. 8). From the perspective of bud categorization of protogynous and protandrous, *JmCOL4a* expression was much higher in protogynous male buds (CX) than protogynous female buds (CC), and the difference in expression fluctuated between 6-10-fold at different times ($p < 0.05$). The expression of *JmCOL4a* in protandrous buds showed the same trend in male and female

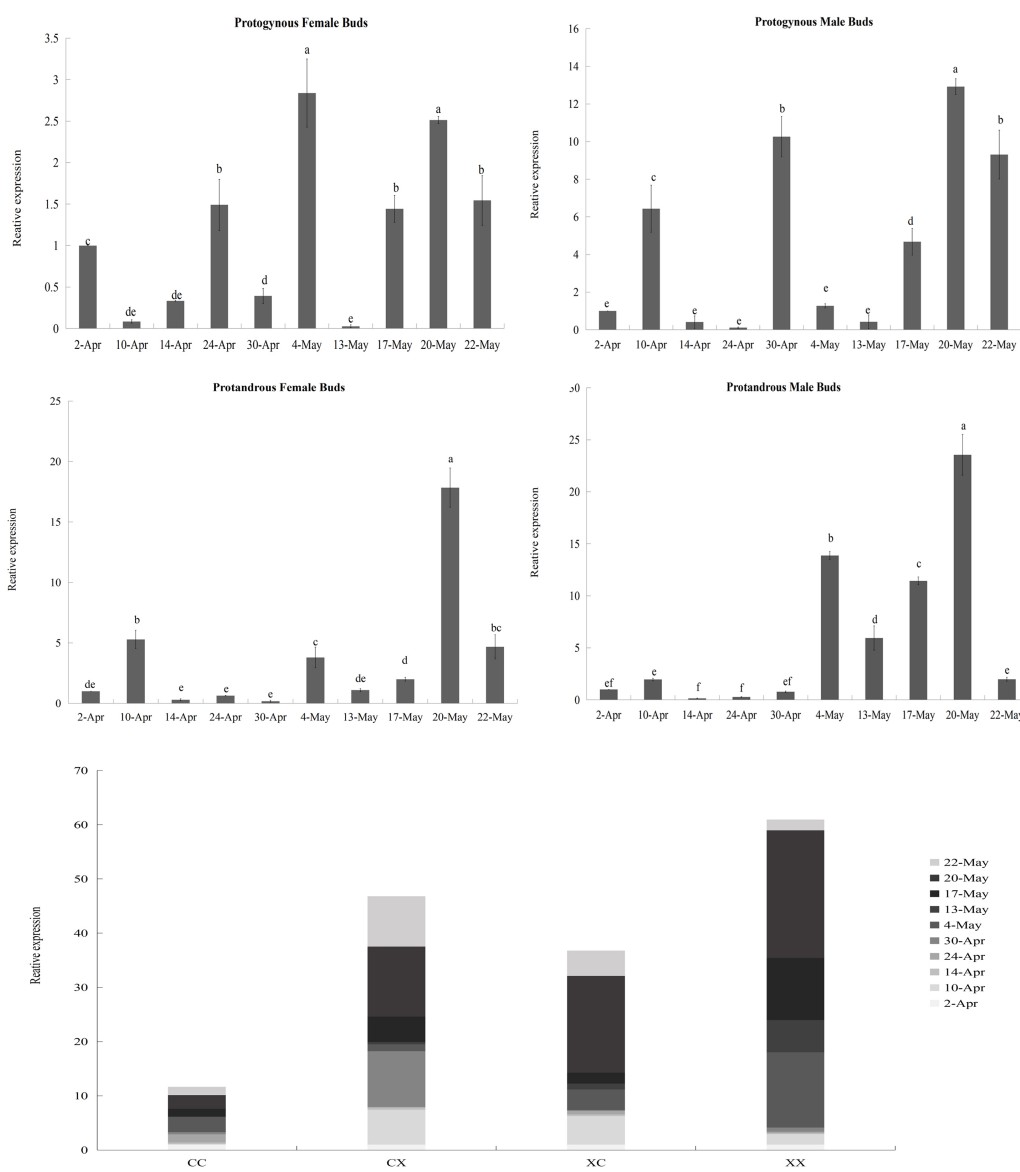

**Figure 8** **Relative expression analysis of *JmCOL4a* in flower buds at different stages.** The flower buds of each period used for analysis were: April 2 (4.2), April 10 (4.10), April 14 (4.14), April 24 (4.24), April 30 (4.30), May 4 (5.4), May 13 (5.13), May 17 (5.17), May 20 (5.20), and May 22 (5.22); the various types of flower buds used for analysis were named: protogynous female buds named CC, protogynous male buds named CX, protandrous female buds named XC, protandrous male buds named XX. The value is the mean ± standard deviation of the three biological replicates. There was a significant difference between different lowercase letters ($p < 0.05$), the same below.

buds, but there was a difference in expression between protandrous male buds (XX) and protandrous female buds (XC), and the difference was mainly reflected on April 10, May 4, May 13 and May 17. Overall, the expression in protandrous (XX) buds was higher than that of protandrous female buds (XC).

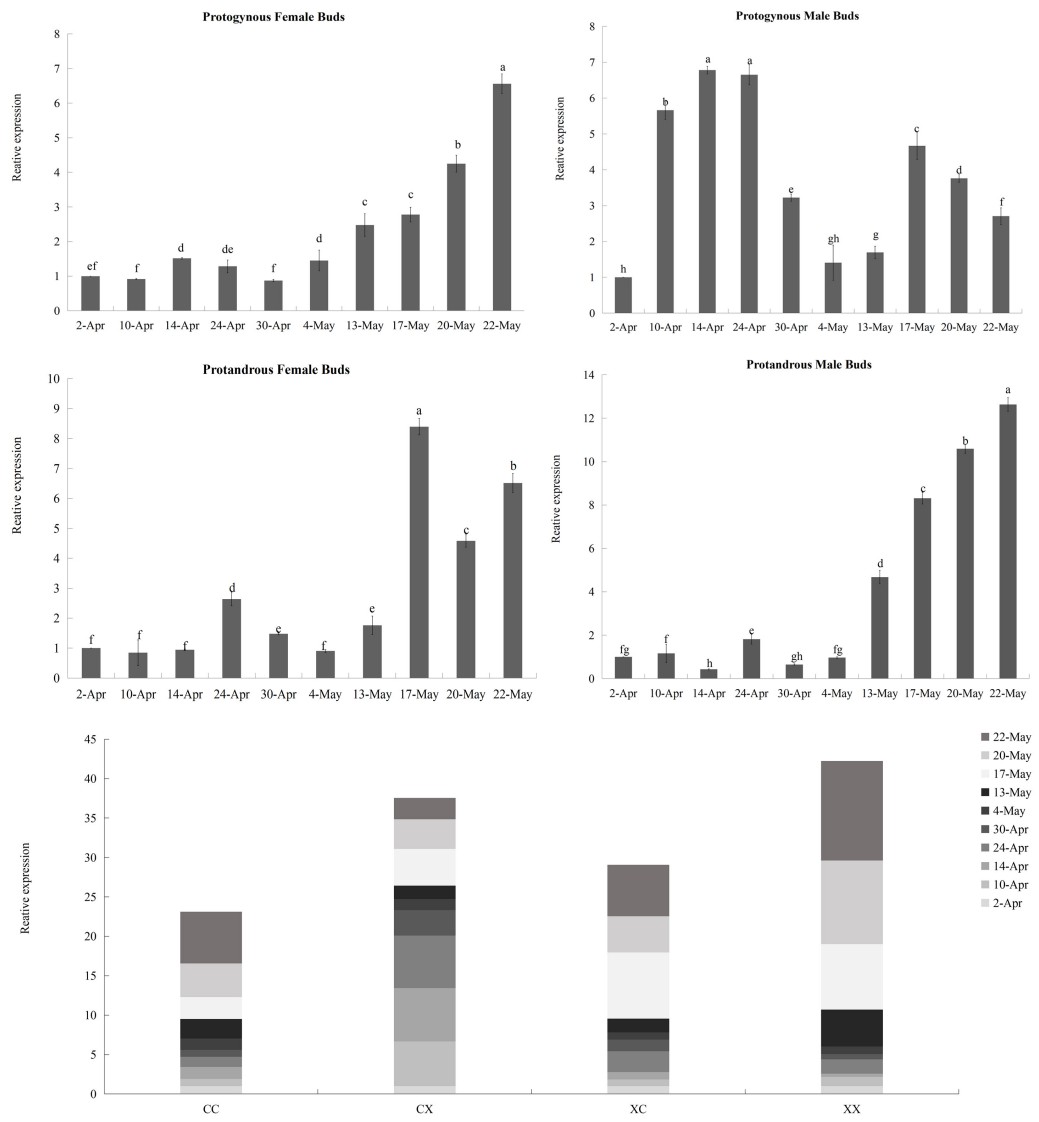

**Figure 9** **Relative expression analysis of *JmCOL5* in flower buds at different stages.** The flower buds of each period used for analysis were: April 2 (4.2), April 10 (4.10), April 14 (4.14), April 24 (4.24), April 30 (4.30), May 4 (5.4), May 13 (5.13), May 17 (5.17), May 20 (5.20), and May 22 (5.22); the various types of flower buds used for analysis were named: protogynous female buds named CC, protogynous male buds named CX, protandrous female buds named XC, protandrous male buds named XX. The value is the mean ± standard deviation of the three biological replicates. There was a significant difference between different lowercase letters ($p < 0.05$).

In contrast to the expression pattern of *JmCOL4a*, at the stage of physiological differentiation (April 2 to April 30) *JmCOL5* in the protogynous male flower buds (CX) appeared to have the highest expression of the four flower buds in this stage, which is in complete agreement with the results of the transcriptome data we measured at the beginning of the physiological differentiation (Table S5). In contrast, by the stage of morphological differentiation (May 4–May 22), *JmCOL5* expression was higher in protandrous than

in protogynous (Fig. 9), and was 2–4 times higher than that in protogynous. The highest value of *JmCOL5* expression in protandrous male flower buds (XX) at different periods was about 30 times higher than the lowest value ($p < 0.05$). *JmCOL5* had different expression patterns in protogynous and protandrous flower buds, and this difference was reflected in protogynous male flower buds (CX). In conclusion, *JmCOL5* expression in these four types of flower buds was highest at the stage of morphological differentiation, except for the female-prior type male flower buds (CX), where the highest value was reached at the stage of physiological differentiation.

## DISCUSSION

Previous studies have shown that a number of key hub genes, including *FT*, *SOC1*, and *CO*, form a complex flowering regulatory network to orchestrate plant flowering (*Zhang et al., 2019*). *CO/FT* regulators play important roles in the photoperiodic regulation of the flowering pathway (*Samach et al., 2000*). *CO* coordinates light and biological clock inputs to induce the expression of the flowering locust (FT) gene, mainly in leaves, which has been verified in the quantitative expression in this paper, where COL family genes are widely expressed in leaves. These proteins are widespread across species, from lower plants such as mosses to algae, which may exhibit a strong photoperiodic response, to higher plants such as monocot and dicot. These transcription factor genes include CO in *A. thaliana*, Hd1 in rice, and its *CO* homologs in *Hordeum vulgare*, *Beta vulgaris*, and *Glycine max* (*Zhang et al., 2015*).

In this study, we identified genes containing both B-box and CCT domains in the *J. mandshurica* genome. The 15 *JmCOLs* genes were mapped to nine chromosomes, suggesting that this gene family is evenly distributed across the chromosomes (Fig. 2). Tandem duplications are characterized by multiple members of a family occurring in the same or adjacent gene spacer regions. The most representative tandem duplicated genes are adjacent homologs on the same chromosome (*Robson et al., 2001*).

A phylogenetic analysis revealed that the *JmCOL* family can be classified into three distinct subgroups, designated as Groups I–III. Concurrently, gene structure analysis demonstrated that genes within the same subgroups exhibit analogous exon and intron structures (Figs. 4A, 4B), suggesting a correlation between evolutionary processes and gene structure. In the present study, motif 1 and motif 2 were identified in all *JmCOL* proteins. Motif 1 and motif 2 encode the CCT and B-box structural domains, respectively. The B-box may be involved in protein interactions, whereas the CCT structural domain is involved in nuclear localization of proteins and interactions with *HAP3* and *HAP5*. Other studies have similarly indicated the crucial function of this motif in regulating transcriptional activity. The *CO* family in *A. thaliana* can be categorized into three main taxa. Taxon 1 includes *CO* and *COL1~COL5*, taxon 2 includes *COL6~COL8* and *COL16*, and taxon 3 includes *COL9~COL15* with one B-box and two evanescent zinc finger structural domains (*Bendix et al., 2013*; *Griffiths et al., 2003b*). The phylogenetic analysis demonstrated that *JmCO2*, *JmCOL4a*, *JmCOL4b*, and *JmCOL5* are situated within the same subclade as *AtCOL1-5*. Moreover, the distribution of *JmCOL* protein motifs is notably specific, with motif 7 being

prevalent only in subgroup II and motifs 4 and 9 being observed exclusively in subgroup III.

*CO* is a light-responsive transcription factor that exhibits a specific circadian expression pattern under LD or SD conditions (*Robson et al., 2001*). It has been demonstrated that COL genes are extensively expressed throughout the life cycle of plant nutrient and reproductive organs, with a particularly high level of expression observed in leaves, which are responsible for sensing external light in plants (*Zhao et al., 2022*). In A. thaliana, the leaf transmits the perceived light signal to CO, which in turn activates the expression of FT and promotes flower formation. For example, *AtCOL8* interacts with *AtCO* to repress the activation of *AtFT* and ultimately delay flowering time. *AtCO* forms a protein complex through its N-terminal B-BOX structural domain and simultaneously recognizes multiple cis-acting elements (CORE1, CORE2, P1, P2) in the *AtFT* promoter (*Wang et al., 2016*). In the present study, the relative expression of nine *COLs* was analyzed in leaves, stems, flower buds, and fruits of *J. mandshurica* by qRT-PCR. The results demonstrated that these genes were widely expressed in different tissues and organs, with notable differences in expression levels across different tissues. However, all nine genes were abundantly expressed in leaves, with *JmCOL4a*, *JmCOL5*, and *JmCOL10* exhibiting particularly high expression levels in flower buds.

A substantial body of research has demonstrated that both *CO* and *COL* genes serve as pivotal regulators in the control of flowering time (*Xu et al., 2022*). In *A. thaliana*, the gene expression of FT is primarily activated by *CONSTANS* (*CO*) (*Luccioni et al., 2019*). While FT proteins are synthesized only in leaves, *FT* is also a critical component of the flowering pathway. It can thus be surmised that the elevated expression of *JmCOLs* genes in leaves may facilitate the activation of *FT* synthesis and thereby promote flowering. *J. mandshurica* reproductive types can be classified into two main categories: protogynous and protandrous, which alternate between male and female flower buds for pollination. This effectively avoids the occurrence of self-fertilization (*Bao et al., 2020*). To investigate the molecular regulation of flower development in *J. mandshurica*, this study employed qRT-PCR to examine the expression patterns of *JmCOL4a* and *JmCOL5* genes in *J. mandshurica*. flower buds at the physiological and morphological differentiation stages. The expression of *JmCOL4a* was found to be significantly differentially expressed in the physiological and morphological differentiation phases of flower bud development in *J. mandshurica*. Furthermore, the expression was significantly higher in male buds than in female buds ($p < 0.05$). The specific expression of *JmCOL5* in the physiological development period of the male flower buds of the protogynous type indicates the potential of participating in the sex differentiation process of the flower buds of *J. mandshurica*. Prior research has demonstrated the existence of two CO genes with specific positive regulatory functions for male inflorescence development in *J. mandshurica* (*Chen et al., 2022*). In light of these findings, we hypothesize that *JmCOL4a* and *JmCOL5* play critical roles in photoperiod-mediated floral development in *J. mandshurica*. In addition to the photoperiodic pathway, these genes also involved in GA biosynthesis and signaling which also play a pivotal role in the regulation of flowering in plants. As corepressors of the GA pathway, DELLAs bind directly to CO through their CCT domain and sequester CO to the FT promoter, thus

directly down-regulating FT and TSF expression at low GA levels. In woody plants, there have been studies revealed that the inhibitory effect of GA on flowering is mediated through *JcFT* and demonstrated the effects of JcGA20ox1, *JcGA2ox6* and *JcFT* on agronomic traits in *Jatropha curcas* (*Li et al., 2022*).

Long-day signaling has great potential applications in agricultural biotechnology for artificially modulating crop photoperiodic response. By altering the photoperiodic response, we can change not only flowering time but also other important traits such as dormancy, growth rate or crop yield (*Huang et al., 2024*). The identification and analysis of the *JmCOLs* gene family can provide a better understanding of the response and potential functions of *J. mandshurica* to long-day signals.

## CONCLUSIONS

In this study, a comprehensive analysis of the *COL* gene family in the *J. mandshurica* genome was conducted, resulting in the identification of a total of 15 *COL* genes. Chromosomal localization analysis revealed that *JmCOLs* were dispersed throughout the *J. mandshurica* chromosomes in a non-clustered manner. Phylogenetic tree analysis indicated that the family was divided into three subfamilies, and that the conserved structural domains and gene structures of the same subfamily were similar. In addition, a large number of light-responsive cis-elements in the promoter region of *JmCOLs* revealed its light-responsive function as a core gene of the photoperiod pathway, which is helpful to further explore the core model of photoperiod regulation of flower development. Tissue-specific expression analysis of *JmCOLs* in different plants revealed the spatiotemporal expression patterns of *JmCOLs* genes, indicating that they play an important role in the life cycle of plant growth and development, and have the potential to participate in the molecular mechanism of photoperiod-induced flower development in *J. mandshurica*. In this study, the *JmCOL4a/5* gene screened by transcriptome data of flower buds at physiological differentiation stage had different expression patterns at different developmental stages of four types of flower buds, which provided a theoretical basis for further exploring the core module of photoperiod regulation of flower development in *J. mandshurica*. In conclusion, we conducted a comprehensive analysis of the structural features and expression patterns of the *JmCOL* gene family. This contributes to a more comprehensive understanding of the biological and molecular functions of the *J. mandshurica JmCOL* gene family and its role in light response.

### Funding

This research was funded by the Applied Basic Research Project of the Liaoning Provincial Department of Science and Technology (2022JH2/101300170). The funders had no role in study design, data collection and analysis, decision to publish, or preparation of the manuscript.

## Grant Disclosures

The following grant information was disclosed by the authors:
Applied Basic Research Project of the Liaoning Provincial Department of Science and Technology: 2022JH2/101300170.

## Competing Interests

The authors declare there are no competing interests.

## Author Contributions

- Jingwen Wu conceived and designed the experiments, performed the experiments, analyzed the data, prepared figures and/or tables, authored or reviewed drafts of the article, and approved the final draft.
- Mengmeng Zhang conceived and designed the experiments, performed the experiments, analyzed the data, prepared figures and/or tables, authored or reviewed drafts of the article, and approved the final draft.
- Yue Gao performed the experiments, prepared figures and/or tables, authored or reviewed drafts of the article, and approved the final draft.
- Shuhan Li performed the experiments, prepared figures and/or tables, authored or reviewed drafts of the article, and approved the final draft.
- Ruoxue Jia performed the experiments, authored or reviewed drafts of the article, and approved the final draft.
- Lijie Zhang conceived and designed the experiments, analyzed the data, prepared figures and/or tables, authored or reviewed drafts of the article, and approved the final draft.

## Data Availability

The data is available at NCBI BioProject: PRJNA356989 and at https://cmb.bnu.edu.cn/juglans.

The raw data are available in the Supplemental Files.

The flower bud transcriptome data is available at NCBI: PRJNA693587; SAMN17392498–SAMN17392509.

## Supplemental Information

Supplemental information for this article can be found online at http://dx.doi.org/10.7717/peerj.19169#supplemental-information.

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
