# Peer review of "Genome-wide characterization and expression analysis of the CONSTANS-like gene family of Juglans mandshurica Maxim"

_PeerJ, doi:10.7717/peerj.19169_

## Round 0.1 · original submission · Major Revisions

Dear Authors

The reviewers have recommended revisions to your manuscript. Therefore, I invite you to respond to the reviewers' comments and revise your manuscript.

In addition, there are significant concerns about the manuscript's grammar, usage, and overall readability. We, therefore, request that you revise the text to fix the grammatical errors and improve the overall readability of the text.

With Thanks

Reviewer 1 ·

Basic reporting

Introduction and background to show context. The literature is well-referenced and relevant.

Experimental design

The research question is well-defined, relevant, and meaningful. It is stated how the research fills an identified knowledge gap.

Validity of the findings

Conclusions are well stated and linked to original research questions.

Additional comments

(1) Professional English language writing should be improved.

(2) The pictures of plant materials were absent in this article; they should be added with the general picture of Juglans mandshurica and the plant materials for RT-PCR. Not all readers familiar with Juglans mandshuric.

(3) lines 33 – 36, gene numbers or specific genes should be displayed in the abstract if they do have a specific expression in leaves, flower buds, or special flower bud development stages.

(4) lines 36 -38, “evolution” is a big word usually used in multiple species evolution and to discuss the difference from ancient to present. “function” usually means genes have a special role in plant organ formation or regulation. The last sentence should be rewritten with an ordinary representation, such as phylogenetic, motif and domain analysis, expression manners, or others. “evolution” and “function” are big and profound words in plant evolution and molecular area.

(5) lines 62-64, the reference should be mentioned. Line 105: the genome and flower bud transcriptome paper should be cited and the project number or the data website links should be mentioned here or in the back of the manuscript.

(6) line 172, Latin names of the sampled plants should be mentioned. In line 206 or others, the Latin name should be mentioned and kept in italic type.

(7) line 191, “The family genes were cloned”, does it mean the authors cloned all 15 COL genes? The genome gene sequence or CDS sequence should be classified, and the primers cloned the family genes could not be found in the supplementary tables. Suppose the family genes (15 COL genes) were only searched from the reported genome database. In that case, the gene ID of the 15 COL genes from the gff3 file of the genome should also be listed on the supplementary tables, the re-ID of the 15 COL genes should correspond to the original ID of the genome, so other readers could find the 15 COL genes sequence from genome data easily.

(8) line 231, where is Table 3? Does it mean Table 2?

(9) line 244 and line 248, where is Figure 4 in the manuscript? The referenced figures should be normalized.

(10) The gene names in the raw data of tissue experiments should be consistent with the gene names in the manuscript or the figures. Some Chinese language in the raw data of tissue experiments could be changed to English based on the journal publication requirements.

(11) Usually, in research papers, one figure would be followed by a table used to draw the figure or a statistic table. The statistic table of Figure 1, Figure 4, or others might be added.

(12) Others might need carefully checked by the authors.

Annotated reviews are not available for download in order to protect the identity of reviewers who chose to remain anonymous.

Reviewer 2 ·

Basic reporting

The manuscript titled "Genome-wide characterization and expression analysis of the CONSTANS-like gene family of Juglans mandshurica Maxim" presents valuable insights into the identification, phylogenetic classification, and expression analysis of the COL gene family in Juglans mandshurica. The study provides a comprehensive genome-wide analysis and explores the potential role of COL genes in photoperiodic regulation of flowering. However, manuscript in its current form needs extensive revision for the correction of typographical, grammatical, and technical errors.
Keywords are missing, provide 5-7 unique keywords.
The introduction is well-structured in terms of background information, but there are abrupt transitions between different topics. For example, the literature jumps from general flowering mechanisms to the specific CO/COL genes without a smooth connection. Some sentences are repetitive, for instance, the regulatory role of CO/COL genes is mentioned multiple times without adding new insights. References are somewhat outdated. The sudden switch to Juglans mandshurica in line 93 feels out of place, as the manuscript has primarily focused on more well-known model plants. Introduce Juglans mandshurica earlier, along with why this study fills a gap in knowledge. Consider discussing the economic and ecological significance of the plant in more detail, providing a stronger justification for studying its flowering mechanisms.
In materials and methods section add clear headings for each subsection. Provide specific details on the locations of sample collection (e.g., geographical coordinates, exact experimental conditions). Clarify the conditions under which the samples were collected (e.g., time of day, weather conditions). Include details about the criteria for selecting candidate proteins. Mention the versions of the software used for analysis. Provide more detail on the "Maximum Likelihood method" (e.g., specify parameters or settings used for analysis). Clarify the sequence alignment criteria and parameters used in MUSCLE. Provide details on the number of samples and the specific method used for RNA extraction, mention all the instruction that authors followed for extraction. Specify the volumes and concentrations used during cDNA synthesis. Clarify how data were analyzed beyond stating that a one-way ANOVA was performed (e.g., which post-hoc test was used).
In Results section, please incorporate detailed statistical analyses, discuss gene expression levels with fold changes in different tissues, and flower buds, and provide deeper interpretations of the results obtained.
In Discussion section, provide in-depth mechanistic exploration on how JmCOL4a and JmCOL5 might function differently in male and female buds that could be bolstered by references to similar mechanisms in other species. Include additional recent studies on sex-specific floral regulation or CO/COL gene function in other woody plants to build a stronger comparison. You might also explore how these findings could be linked to epigenetic or post-translational modifications, which are emerging as important regulatory layers in plant reproduction. The interaction between photoperiod and gibberellin (GA) signalling is mentioned, but this point needs further clarification. Expand the discussion on GA signalling and its intersection with CO/COL regulation in Juglans mandshurica, possibly citing recent findings on GA’s role in regulating flowering time in other trees.
The Conclusion section effectively summarises the key findings but could be more concise. Some points are repetitive and can be condensed for better readability. The section highlights the main discoveries about JmCOL genes, but it lacks future perspective on how this research could be applied or further explored. Add a forward-looking statement on the potential applications of this research.
Please ensure that the in-text citations and the reference list adhere to the journal's required style. Additionally, many of the references are missing DOIs. Kindly include the DOI for each reference where available to meet the journal's standards.
In Figures 5–7, the letter assignment is incorrect in several sub-figures, as in in Figure 5 (JmCOL9a). The mis-assignment of letters may lead to misinterpretation of the data, compromising the accuracy of the statistical analysis. I recommend thoroughly reviewing and correcting the letter assignments to ensure they accurately reflect the significant differences between treatments.
Please revise the footnote of Figure 5 for clarity. Specifically, remove the phrase "the same below" and instead provide a separate footnote for Figures 6 and 7. The revised footnote for Figure 5 should read: “The tissue development stages used for analysis were: S (stem), FB (flower bud), F (fruit), L (leaf). Values represent the mean ± standard deviation of three biological replicates. Different lowercase letters indicate significant differences between treatments (p<0.05), the same below” Ensure that the footnotes for Figures 6 and 7 similarly specify the statistical comparison without referring to previous figures.
Specific Comments:
Lines 34-36: the statement "Furthermore, significant differences (p<0.05) in expression were noted at different stages of flower bud development" should be more precise. It is recommended to provide exact values representing the percentage increase or decrease in expression levels, rather than just stating the significance. This will offer a clearer and more quantitative understanding of the changes observed.
Line 51-52: "contain one or two B-box structural domains at the N-terminus. These domains consist of two cysteines..." should specify zinc finger B-box domains to improve accuracy.
Line 72-74: "In contrast, AtCOL3 and AtCOL4 were identified as flowering repressors..." This section is somewhat unclear regarding how repressors function in both LD and SD conditions. Clarify how this differs mechanistically.
Ensure consistency in naming conventions for genes and proteins (e.g., "JmCOL" should be consistently styled).
Line No. 115-116: Consider rephrasing for clarity. For example, "Ten samples of gynogenic and androgenic flower buds from Juglans mandshurica were collected from various developmental stages on the following dates: April 2 (4.2), April 10 (4.10), April 14 (4.14), April 24 (4.24), April 30 (4.30), May 4 (5.4), May 13 (5.13), May 17 (5.17), May 20 (5.20), and May 22 (5.22)." Remove dates like 4.2, 4.10.
Lines 161-164: Specify which version of the genome data was used and provide the source if applicable.
Lines 165-170: Describe how the 2 kb upstream region was defined (e.g., based on genome annotation).
Lines 362-366: The qRT-PCR data showing the expression patterns of JmCOL genes in different tissues is a strong point but lacks functional interpretation. Offer a clearer mechanistic explanation for how high JmCOL expression in leaves could influence flowering via FT regulation. Relating this to the characterised Arabidopsis CO-FT pathway would strengthen the argument, particularly when interpreting why certain JmCOL genes are highly expressed in flower buds.
Line 325: "screened for genes containing both B-box and CCT structural domains" could be phrased more precisely as "identified genes containing both B-box and CCT domains in the Juglans mandshurica genome."
Line 379: "we further postulate that JmCOL4a and JmCOL5 may also be implicated in the photoperiod-mediated flowering process" would be clearer as "we hypothesize that JmCOL4a and JmCOL5 play critical roles in photoperiod-mediated floral development."
Lines 403-407: Several points in the conclusion are repetitive from the Discussion. For example, about the spatiotemporal expression patterns of JmCOL genes, condense this section by avoiding repetition of details already discussed. Focus on the most significant takeaways rather than reiterating all findings.

Experimental design

Experimental design needs clarification as mentioned in basic reporting.

Validity of the findings

The validity of the findings is currently undermined by the lack of precise statistical reporting and insufficient detail on the extent of expression differences. Providing exact quantitative data, along with robust statistical analyses, is essential to ensure the credibility and reproducibility of the study's results.

Reviewer 3 ·

Basic reporting

This study identified 15 JmCOLs family genes in Juglans mandshurica. Phylogenetic analysis classified them into three subgroups. Chromosome analysis showed their distribution. Multiple sequence alignment and promoter prediction analysis revealed their characteristics. Quantitative real-time PCR analysis demonstrated their tissue-specific expression. The study reveals the molecular evolution and function of JmCOLs and their role in flower bud development.

Experimental design

The author needs to conduct experiments or some bioinformatics analyses, such as transcriptomic analysis of flower development in Juglans mandshurica, to further illustrate that the screened genes have a close relationship with flower development. It is difficult to prove this only through qPCR.

Validity of the findings

This article should have identified several important genes through extensive literature review instead of presenting many details. This way, the significance of the entire article cannot be fully demonstrated. In the introduction, results, and discussion sections of the full text, several candidate genes should be carefully selected and highlighted.

---

## Round 0.2 · Major Revisions

Dear Authors

The reviewers have recommended revisions to your manuscript. Therefore, I invite you to respond to the reviewers’ comments and revise your manuscript.
In addition, there are significant concerns about the manuscript's grammar, usage, and overall readability. We, therefore, request that you revise the text to fix the grammatical errors and improve its overall readability.

With Thanks

Reviewer 1 ·

Basic reporting

The article must be written in English and grammar checked.
In the merged PDF, many citations showed “Error! Reference source not found.” Authors should check all the citations before submitting to the journal to avoid obvious writing and format errors.
Key citations should be added, such as lines 83-86, 93-97, and others.
The format of the citations should be normalized, where some were with the author and years, and where some were with Arabic numbers.
Only supplementary material Fig. 1 and Supplementary Table 5 were cited and mentioned in the manuscript.
Remove the file of Supplymental_Table_1.xlsx.
The primers for RT-PCR were still missing.

Experimental design

no comment.

Validity of the findings

The sentence in line 129 “Data were collected as previously described in Yang et al. (2024)” was unclear and ambiguous.
Line 188, “The project number is YFnj20201682”, the name or link of the submitted database should be added, such as NCBI, or others.
Line 206, as the genome database (https://cmb.bnu.edu.cn/juglans/) shows two versions of the genome released, the assembled version used in this manuscript should be mentioned(V1.3 or V2.0).

Additional comments

Some sentences were repeated and redundancy with so many details in the abstract, it should keep the main content and key results, such as the 500-word (3,000 characters) limit of the abstract.
Lines 143-145, line 190, and others should be checked and reorganized.
Line 166 and Line 184, are the sentences described correctly, check again.
Line 224, “The family genes were cloned”, if not cloned all the sequences, remove this sentence.  
Line 281 “19 pairs of genes exhibited homology…” Is there any statistic table belonging to Figure 5?
Line 468, “rowan genome”, is it means the genome of Juglans mandshurica.
Line 191, in my opinion, is not a good choice to name ten periods as 1-10 in order because each period included four different flower types, thus, using the date to represent the ten developing periods might be acceptable: April 2, April 10,……
Line 148 and Line 153, Arabidopsis includes many species. A. thaliana can be used after you have mentioned Arabidopsis thaliana, the same as other plant species.
Other manuscript requirements depend on the author's instructions for this journal (https://peerj.com/about/author-instructions/ ).

Reviewer 2 ·

Basic reporting

Although the authors have attempted to address most of the suggestions, the article still contains numerous typographical, grammatical, and technical errors that require carefull attention. Additionally, several sentences are excessively long and include unnecessary repetitions throughout the manuscript.
In introduction the text is frequently repeating some ideas, such as the role of the CONSTANS (CO) gene in photoperiodic flowering and COL gene conservation across plants (e.g., the same concept is discussed multiple times in different contexts
The extensive list of COL gene members across various species (lines 68–73) and their exact counts must be summarized, focusing on the diversity of COL genes across plant species and their evolutionary significance.
Multiple instances of "Error! Reference source not found." undermine the document's credibility and disrupt the flow. Correct citation errors and ensure all references are consistent with journal guidelines.
Refine the objectives by explicitly linking them to the potential implications for agriculture, forestry, or conservation.
Line 202: "Appropriate number of samples were collected..." Specify the number of replicates and the rationale for their selection to ensure statistical robustness.
Figures 8 and 9 still have incorrect lettering, and their axis titles also contain typographical errors.

Experimental design

The experimental design is satisfactory; however, the statistical analysis presented in Figures 8 and 9 must be repeated, as the lettering in the figures is incorrect and requires revision.

Validity of the findings

The validity of the findings is contingent upon the implementation of appropriate statistical analysis.

Reviewer 3 ·

Basic reporting

All the entries have been modified very well, there is no other opinion.

Experimental design

Experimental design are well.

Validity of the findings

The conclusions are well.

---

## Round 0.3 · Minor Revisions

Dear Authors
The manuscript still needs a minor revision before publication. The authors are invited to revise the paper considering all the suggestions made by the reviewers. Please note that the requested changes are required for publication.
With Thanks

Reviewer 1 ·

Basic reporting

no comment

Experimental design

no comment

Validity of the findings

no comment

Additional comments

(1) Table 1: The primer sequence of actin (internal reference) for qPCR was missing.
(2) Table 2: The information in Table 2 about the qPCR instrument, procedures, and reaction conditions might be suitably merged into the methods section.
(3) Table 3 and Table 5: Column names should be added, checked, and carefully considered. Research paper readers might be from all over the world, so ensure all tables are readable and easily understood by others.
(4) Arabidopsis is a genus including many plant species.
(5)Why not directly use April 2, April 10, April 14, April 24, April 30, May 4, May 13, May 17, May 20, and May 22 on behalf of ten periods? Such as lines 189-190,  Figure 8-9, and the content in tables for Figure 8-9.  Additionally, the bar chat might be adjusted to be more visualized.
(6) Other content might need authors to check the manuscript thoroughly, and details depend on the author guidelines of the journal.

Reviewer 2 ·

Basic reporting

The manuscript still contains multiple grammatical errors, broken citations, and inconsistencies in software names, website URLs, and terminology. To enhance clarity and readability, I strongly recommend consulting a professional editor or a native English speaker for thorough language and formatting revision.
Ensure proper formatting of all references (e.g., avoid double citations like [15][15]).
Use precise terms (e.g., Neighbor-Joining (NJ) tree, p-distance model), and correct units where necessary (e.g., -80°C without unnecessary spacing).
Unfortunately, the previously identified issue persists in the manuscript. The placeholder error "Error! Reference source not found" is still present in Line 75 and Line 420. Please ensure that all references are correctly linked and properly formatted before resubmission.
The error bars in Figure 8 appear excessively large, which may indicate potential issues with data variability or calculation. Please verify the underlying data, statistical analysis, and replication consistency to ensure accuracy.

Experimental design

Experimental design seems ok.

Validity of the findings

Findings of Figure 8 are ambiguous on account too enlarged standard errors bars. Please recheck it.

---

## Round 0.4 · Minor Revisions

Dear Authors
The manuscript still needs a minor revision before publication. The authors are invited to revise the paper considering all the suggestions made by the reviewers. Please note that the requested changes are required for publication.
With Thanks

Reviewer 1 ·

Basic reporting

no comment

Experimental design

no comment

Validity of the findings

no comment

Additional comments

(1) line 285-286, "a total of 19 pairs of genes exhibited homology.....(Figure 5B)", please recheck the 19 pairs list of genes.
(2) the title of Figure 9 was repeated with Figure 8, please recheck the Figure legend.
(3) other comments have been mentioned in previous versions.

Reviewer 2 ·

Basic reporting

Authors have addressed most of the queries raised in the previous round of revision. Now the article can be accepted for publication in PeerJ.

Experimental design

Good

Validity of the findings

Good

---

## Round 0.5 · accepted · Accept

Dear Authors,

I am pleased to inform you that the manuscript has improved and can be accepted for publication.

Congratulations on accepting your manuscript, and thank you for your interest in submitting your work to PeerJ.

With Thanks